# LANE: Learning with Noisy Labels Using Label-Aware Margins

**Tiberiu Sosea**     **Cornelia Caragea**
Computer Science, University of Illinois Chicago

tsosea2@uic.edu   cornelia@uic.edu

## Abstract

In this paper, we propose Label-Aware Noise Elimination (LANE), a new approach to learning with noisy labels. At its core, LANE introduces a new metric—label-aware margin—aimed at quantifying the degree of noise of each training example (or quality thereof). LANE leverages the semantic relations between classes and monitors the training dynamics of the model on each training example to dynamically lower the weight of training examples that are perceived to have noisy labels. We test the effectiveness of LANE on multiple text classification tasks and benchmark our approach on a wide variety of datasets with various numbers of classes and amounts of label noise. LANE considerably outperforms strong baselines on all datasets and settings, obtaining significant improvements ranging from an average improvement of $2.88\%$ in F1 on manually annotated datasets to a considerable average improvement of $4.75\%$ F1 on datasets with high level of injected label noise. We carry out a comprehensive analysis of LANE and identify the key components that lead to its success. We release our code at: https://github.com/tsosea2/Label-Aware-Noise-Elimination.

## 1 Introduction

Supervised deep learning models are ubiquitous in many applications, but their success depends on the quality of the training data. Many existing datasets are annotated by humans on crowdsourcing platforms (Demszky et al., 2020) or by automatic approaches such as distant (or weak) supervision (Mintz et al., 2009; Wang et al., 2012; Abdul-Mageed & Ungar, 2017), and, while weak supervision inherently introduces unwanted mislabeled samples, humans—no matter how careful, are also prone to making labeling errors, especially on tasks that involve distinguishing between a large number of closely confusable or overlapping classes, e.g., emotion detection (Demszky et al., 2020) or topic classification (Lewis et al., 2004). The mislabeled training samples are particularly harmful when learning large overparameterized neural networks, since these networks can achieve zero training error on any dataset, and have very poor generalization capabilities (Zhang et al., 2016).

Pleiss et al. (2020) proposed the Area Under the Margin (AUM) as a metric to identify mislabeled samples from a training set. The AUM for a sample measures the difference between the logit corresponding to its *assigned* label and the largest logit among all *non-assigned* labels averaged across the training epochs. The *assigned* label is the label assigned by either humans, weak supervision, or even Large Language Models. The AUM for a *mislabeled sample* is expected to be low (likely negative) since the model—through generalization—tends to predict the sample in its (hidden) true class, and hence, the largest logit (among all logits) does not correspond to the assigned (wrong) label, but to the (hidden) true label. Pleiss et al. (2020) subsequently removed samples with low AUM from the training set using a fixed AUM threshold. Similarly, Han et al. (2018); Li et al. (2020) proposed the small-loss trick that uses the loss value in the convergence to identify mislabeled samples as those with a large-loss and removed the large-loss samples from the training set. However, through either the fixed AUM threshold or large loss elimination, hard but valuable clean samples are unnecessarily removed from the training set. More recently, Zhang et al. (2024) proposed to use all training samples, each with a different weight estimated using a sample weighting mechanism called Hyperspherical Margin Weighting (HMW). That is, HMW assigns a weight to each sample according to the Integrated Area Margin (IAM)—an extension of the AUM

metric that contrasts the logit of the assigned label with the two largest other logits. However, neither AUM nor IAM captures any semantic similarities that inherently exist between labels. For example, in emotion detection, "anger" is semantically more similar to "fear" than it is to "joy", and hence, a sample with the true (hidden) label "anger" but with assigned label "fear" should be penalized less than the same sample having the assigned label "joy".

To this end, we introduce **L**abel-**A**ware **N**oise **E**limination (LANE), a new approach to learning with noisy labels that specifically captures semantic relations between labels. In our approach, we retain *all* training samples, but we weight them differently based on the model's behavior on each sample. Thus, similar to Zhang et al. (2024), our model has access to a much larger diversity of samples during training, including the hard but clean ones. In weighting the samples, we estimate the degree of "noisiness" of the assigned labels by introducing *label-aware margins* averaged across training iterations that capture inter-class semantic similarities. Our label-aware margins extend the concept of *margins* (Pleiss et al., 2020) by adaptively weighting samples when the assigned label does not consistently match the model's predicted label (over the training iterations). Note that the model's predicted label likely corresponds to the true (hidden) label when that label is consistently predicted by the model over the training iterations because of the ability of the model to generalize from other training samples that belong to the same label. LANE leverages AUM and jointly trains two networks to identify mislabeled samples and consequently assign a per sample weight that accounts for the semantic similarity between labels so that an assigned mislabel receives a lower weight when it is more distant from the true (hidden) label and a higher weight when it is close to the true label. We learn the inter-class semantic similarities using a label-aware supervised contrastive loss, trained jointly with a cross-entropy loss, to better distinguish between easily confusable labels.

We evaluate the effectiveness of LANE on ten datasets: Empathetic Dialogues (Rashkin et al., 2019), GoEmotions (Demszky et al., 2020), ISEAR (Scherer & Wallbott, 1994), CancerEMO (Sosea & Caragea, 2020), RCV1 (Lewis et al., 2004), SciHTC (Sadat & Caragea, 2022), SST-5 (Socher et al., 2013a), Amazon Review (McAuley & Leskovec, 2013), Yelp Review (Asghar, 2016), and Yahoo Answer (Chang et al., 2008). Using these datasets, we show that LANE works well on a wide range of tasks and domains (emotion and general text classification; social networks, dialogues, and personal experiences). In all our experiments, automatically scaling down the weight of identified noisy samples from the training set shows great potential, improving the average performance on our original datasets by $2.88\%$ F1 over AUM and by $2.32\%$ over HMW. On noisy datasets, our method boosts the performance by an average $3.37\%$ F1 over AUM and $3.4\%$ over HMW. The performance gap is even more pronounced in extreme scenarios; for instance, at $40\%$ label noise, LANE significantly outperforms AUM and HMW by $4.75\%$ and $4.01\%$ F1, respectively.

We summarize our contributions as follows: **1)** We introduce LANE, a new approach that allows models to learn under label noise from a large diversity of samples and, at the same time, leverages inter-class semantic similarities to automatically identify and minimize the harmful effects of noisy samples; **2)** We evaluate the effectiveness of our approach on ten text classification datasets from different tasks and domains and show improvements in performance compared with strong baselines and prior works; **3)** We carry out a comprehensive analysis and ablation study of LANE to validate the effectiveness of our proposed method.

## 2 RELATED WORK

**Noise-robustness** Robustness to noisy labels is addressed through robust loss functions, label correction, and regularization. *Robust loss functions* are designed to mitigate noise and reduce overfitting (Liu & Tao, 2015; Goldberger & Ben-Reuven, 2016; Lin et al., 2017; Ren et al., 2018; Saxena et al., 2019; Wang et al., 2019; Liu & Guo, 2020; Englesson & Azizpour, 2021; Li et al., 2021; Bai et al., 2022; Gao et al., 2023; Garg et al., 2023; Wei et al., 2023a;b;c). For example, Goldberger & Ben-Reuven (2016) propose to add a noise layer in the neural network architecture, whose parameters can be learned for an accurate label estimation. Saxena et al. (2019) introduce a curriculum-learning approach that uses learnable data parameters to rank the importance of examples in the learning process. These parameters are then leveraged to decide the data to use at different training stages. Wang et al. (2019) introduce Symmetric cross entropy Learning (SL), a method that addresses the issue of both under-learning of "hard" classes and overfitting of "easy" classes. Focal loss (Lin et al., 2017) incorporates a soft weighting scheme that puts emphasis on harder samples.

Liu & Guo (2020) on the other hand propose to alter the loss function to make it more robust under label noise and introduce Peer Loss Functions, which evaluate predictions on both the samples at hand, as well as carefully automatically constructed *peer* samples. In our work, we also alter the loss and introduce a weighted cross-entropy loss where a sample's weight reflects its quality or level of noise that is learned jointly using a label-aware supervised contrastive loss. *Label correction* approaches gradually update the assigned labels based on the model's predictions (Tanaka et al., 2018; Yi & Wu, 2019; Li et al., 2022; Xu et al., 2025). For example, Xu et al. (2025) propose a label correction method that uses normalized Jensen–Shannon divergence to dynamically interpolate between noisy labels and model predictions. *Regularization* approaches aim to prevent training data memorization and include early-learning regularization (Liu et al., 2020), mixup (Zhang et al., 2018), label smoothing (Wei et al., 2022), and representation regularizer (Cheng et al., 2023).

**Contrastive learning** Supervised contrastive learning brings the latent representations of input samples closer together if they belong to the same class (*positives*) and further apart if they belong to different classes (*negatives*). Gunel et al. (2020) use a supervised contrastive loss to improve fine-tuning performance of pre-trained language models in few-shot scenarios. Khosla et al. (2020) introduce a variation of the traditional contrastive loss which aims to produce more samples in the *positive* set by using data augmentation to generate more positive samples. Suresh & Ong (2021) argue that not all negative samples are equal and propose label-aware contrastive loss to infer the relations between classes and weight samples differently. Liu et al. (2025) propose a contrastive framework to construct a reliable negative set of each sample by filtering out inappropriate negative pairs. In contrast, we propose *label-aware margins* that adaptively weight samples based on noise levels and inter-class semantic similarity, learned via a label-aware supervised contrastive loss.

**Sample selection** Pleiss et al. (2020) use Area Under the Margin (AUM) to monitor the behavior of the model on each sample and identify low-AUM samples as mislabeled samples. Zhang et al. (2024) propose Hyperspherical Margin Weighting (HMW) to assign a per sample weight based on Integrated Area Margin (IAM). However, neither AUM nor IAM captures the semantic relation between the assigned (wrong) label and the true (hidden) label. Pan et al. (2025) identify correctly labeled yet hard-to-learn samples using the confidence gap between the annotated label and the remaining labels during training to separate clean and mislabeled samples. Swayamdipta et al. (2020) separate training data into three regions, easy-to-learn, ambiguous, and hard-to-learn (potentially mislabeled) and use each region separately (by removing the other two regions) to understand its benefits to learning and generalization. DISC (Li et al., 2023) uses an instance-specific dynamic thresholding to block access to specific training samples based on the momentum of each sample memorization strength. Co-teaching (Han et al., 2018) uses two networks to combate noisy labels, with each network extracting samples with small loss and feeding them to its peer network for further training. DivideMix (Li et al., 2020) divides the training data into a labeled set with clean samples and an unlabeled set with noisy samples and trains the model in a semi-supervised fashion, maintaining two diverged networks where each network uses the dataset division from the other network. Unicon (Karim et al., 2022) considers potentially noisy labeled data as unlabeled samples in a leverages semi-supervised learning framework. We compare LANE with many of the above works.

## 3 PROPOSED APPROACH

Here, we first provide background on Area Under the Margin (AUM) introduced by Pleiss et al. (2020) (§3.1) and then present **L**abel-**A**ware **N**oise **E**limination (LANE), our new approach that leverages AUM to improve model robustness from noisy labels (§3.2).

### 3.1 BACKGROUND

The margin M (Pleiss et al., 2020; Bartlett et al., 2017; Elsayed et al., 2018; Jiang et al., 2018) of a sample $\mathbf{x}$ with assigned label $y$ at a training epoch $t$ is defined as follows:

$$\mathbf{M}^{(t)}(\mathbf{x}, y) = z_y^{(t)}(\mathbf{x}) - max_{k!=y} z_k^{(t)}(\mathbf{x}) \tag{1}$$

where $z_y^{(t)}(\mathbf{x})$ is the logit corresponding to assigned label $y$, and $max_{k!=y} z_k^{(t)}(\mathbf{x})$ is the largest *other* logit corresponding to label $k$ (from among all non-assigned labels). The margin measures how

---

**Algorithm 1** LANE: Label-Aware Noise Elimination

---

1: **Input:** Training data $D = \{(\mathbf{x}_i, y_i)\}_{i=1}^N$, classifier network $\theta$, auxiliary network $\Pi$, total epochs $T$.
2: **Initialize:** For each example $(\mathbf{x}_i, y_i)$, initialize $\text{ALM}^{(0)}(\mathbf{x}_i, y_i) = 0$.
3: **for** epoch $t = 1, \dots, T$ **do**
4:      **for** each batch $B \subset D$ **do**
5:          For each $\mathbf{x}_i \in B$, compute logits $z_k^{(t)}(\mathbf{x}_i)$ using $\theta$ and class weights (probabilities) $w_{\mathbf{x}_i, k}$ using $\Pi$ for all labels $k$.
6:          **for all** $(\mathbf{x}_i, y_i)$ in batch $B$ **do**
7:              $\mathbf{M}^{(t)}(\mathbf{x}_i, y_i) \leftarrow z_{y_i}^{(t)}(\mathbf{x}_i) - \max_{k \neq y_i} z_k^{(t)}(\mathbf{x}_i)$ (Equation 1)
8:              $\text{LM}^{(t)}(\mathbf{x}_i, y_i) \leftarrow \frac{1}{w_{\mathbf{x}_i, j}} \cdot \mathbf{M}^{(t)}(\mathbf{x}_i, y_i)$ **if** $\mathbf{M}^{(t)}(\mathbf{x}_i, y_i) < 0$ **else** $\text{LM}^{(t)}(\mathbf{x}_i, y_i) \leftarrow$
             $\mathbf{M}^{(t)}(\mathbf{x}_i, y_i)$ where $j = \text{argmax}_{k! = y_i} z_k^{(t)}(\mathbf{x}_i)$
9:              $\text{ALM}^{(t)}(\mathbf{x}_i, y_i) = \frac{1}{t} \sum_{r=1}^t \text{LM}^{(r)}(\mathbf{x}_i, y_i)$
10:          **end for**
11:          $N^t \leftarrow \{(\mathbf{x}_i, y_i) \in B \mid \text{ALM}^{(t)}(\mathbf{x}_i, y_i) < 0\}$
12:          Compute $\mu_t$ and $\sigma_t^2$ according to Equations 5 and 6.
13:          **for all** $(\mathbf{x}_i, y_i)$ in batch $B$ **do**
14:              $\lambda_{CE}^t(\mathbf{x}_i, y_i) \leftarrow 1$
15:              **if** $(\mathbf{x}_i, y_i) \in N^t$ and $\text{ALM}^{(t)}(\mathbf{x}_i, y_i) < \mu_t$ **then**
16:              $\lambda_{CE}^t(\mathbf{x}_i, y_i) \leftarrow \exp\left(-\frac{(\text{ALM}^{(t)}(\mathbf{x}_i, y_i) - \mu_t)^2}{2\sigma_t^2}\right)$
17:              **end if**
18:          **end for**
19:          $\mathcal{L}_{LSCL} = \sum_{i=1}^{|B|} H(\Pi(\mathbf{x}_i), y_i) + \sum_{i=1}^{|B|} \frac{-1}{|P_{\mathbf{x}_i}|} \sum_{p \in P_{\mathbf{x}_i}} \log \frac{w_{\mathbf{x}_i, y_{\mathbf{x}_i}} \cdot \exp(h_{\mathbf{x}_i}^\theta \cdot h_p^\theta)}{\sum_{s \in B; y_s \neq y_{\mathbf{x}_i}} w_{\mathbf{x}_i, y_s} \cdot \exp(h_{\mathbf{x}_i}^\theta \cdot h_s^\theta)}$
20:          $\mathcal{L}_{wCE} \leftarrow \sum_{i=1}^{|B|} \lambda_{CE}^t(\mathbf{x}_i, y_i) \cdot H(\theta(\mathbf{x}_i), y_i)$
21:          Minimize $\mathcal{L} \leftarrow \mathcal{L}_{wCE} + \mathcal{L}_{LSCL}$
22:      **end for**
23: **end for**

---

different the assigned label is compared to a model's *belief* in a label at some epoch. A negative margin likely implies an incorrect prediction, whereas a positive margin implies a correct prediction. The label quality (or noise) of a sample $\mathbf{x}$ is measured by averaging the margins of $\mathbf{x}$ across all training epochs T, i.e., the Area Under the Margin (AUM) (Pleiss et al., 2020), defined as follows:

$$\text{AUM}(\mathbf{x}, y) = \frac{1}{T} \sum_{t=1}^{T} \mathbf{M}^{(t)}(\mathbf{x}, y) \tag{2}$$

Pleiss et al. (2020) first identify mislabeled samples by learning a threshold of separation between the AUMs of clean and erroneous samples through a new artificial class that mimics the training dynamics of mislabeled data and then remove all samples that fall under this threshold.

## 3.2 Our Proposal: Label-Aware Noise Elimination

While the AUM metric is effective for identifying noisy data, it has two key weaknesses: 1) it treats all label errors equally, ignoring the semantic relations between classes, and 2) it relies on a hard threshold to completely remove samples, which can discard valuable but difficult clean samples. To address these issues, we introduce Label-Aware Noise Elimination (LANE). Instead of removing samples, LANE retains all training samples and assigns a per sample dynamic weight. This is achieved by jointly training two networks to assess not only the likelihood of a mislabel but also its semantic severity, ensuring that hard-but-clean samples are preserved while the impact of noisy labels is minimized. The core of this mechanism is a redefinition of the traditional margin. We call this new metric the Label-aware Margin (LM). Algorithm 1 presents the learning of LANE.

**Label-aware Margin (LM)** LM operates within LANE's two-network architecture, where a main classifier network $\theta$ and an auxiliary network $\Pi$ are trained jointly. The LM rescales the standard margin M—calculated from the logits of the main classifier $\theta$—now using semantic similarity weights produced by the auxiliary network $\Pi$. This rescaling is applied specifically when the margin is negative, which is a strong indicator of a mislabel. We adjust the margin dynamically as follows:

| TEXT | | SDN | JOY | FER | ANG | SRP | DSG | TRS | ANT | M | LM |
|---|---|---|---|---|---|---|---|---|---|---|---|
| | | | | | | LOGITS | | | | | |
| $\mathbf{x}_1$ | The doctors do not have any options for him. | 1.1 | 0.45 | **1.2** | **1.8** | 0.27 | 1.56 | 0.11 | −0.7 | −0.6 | −0.67 |
| $\mathbf{x}_2$ | I have found so much info and support on this site, and yet they accept me for who I am. | 1.1 | 1.56 | **1.2** | 0.45 | 0.27 | 0.11 | **1.8** | −0.7 | −0.6 | −1.15 |

Table 1: Comparison of Margin (M) and Label-aware Margin (LM) for two examples. The assigned label (fear) is shown in **red bold** and the model predicted label for each example is shown in **blue bold**. For both examples, we observe that M is $-0.6$ (i.e., $1.2 - 1.8$). In the first example, LM is rescaled slightly since the assigned emotion fear is semantically close to the emotion corresponding to the largest other logit (i.e., anger). In contrast, we observe that in the second example, the assigned emotion fear is semantically distant from the emotion corresponding to the largest other logit which is trust, and hence, LM becomes much smaller.

$$\text{LM}^{(t)}(\mathbf{x}, y) = \begin{cases} \frac{1}{w_{\mathbf{x},j}} \cdot \text{M}^{(t)}(\mathbf{x}, y) \text{ if } \text{M}^{(t)}(\mathbf{x}, y) < 0 \\ \qquad \text{where } j = \text{argmax}_{k! = y} z_k^{(t)}(\mathbf{x}) \\ \text{M}^{(t)}(\mathbf{x}, y) \quad \text{otherwise} \end{cases} \tag{3}$$

where $w_{\mathbf{x},j}$ is the weight obtained using the auxiliary network $\Pi$, which produces higher values if the (potentially wrong) assigned label $y$ of $\mathbf{x}$ is semantically close to the (hidden) likely true label $j$ predicted by the model, and lower values otherwise (i.e., if the potentially wrong assigned label is semantically distant from the model prediction). Note that we scale the margins only if the margins are negative, since these are the potentially problematic samples that may be overly ambiguous or mislabeled. To showcase the difference between our proposed label-aware margin LM and the vanilla margin M, we present in Table 1 two examples from an emotion dataset alongside the logits produced by the model as well as the margin M and label-aware margin LM. Both of these examples have the assigned label the *fear* emotion—while $\mathbf{x}_1$ can be viewed as ambiguous, $\mathbf{x}_2$ is clearly mislabeled. However, although the margin of both examples is the same $\text{M} = -0.6$, we notice that the assigned label fear is semantically close to the label corresponding to the largest other logit (i.e., anger)—the model prediction in the first example, whereas in the second example, it is semantically distant from the label corresponding to the largest other logit (i.e., trust)—the model prediction. We emphasize that our LM captures this semantic difference between labels. Specifically, we observe that the LM of the first example, where the prediction and the assigned label are semantically close, i.e., anger and fear, is larger than the LM of the second example where the prediction and the assigned label are semantically distant, i.e., trust and fear.

**Average Label-aware Margin (ALM)** At an arbitrary iteration $t$ we average the LMs across the training iterations, from the beginning up until the current iteration $t$ and obtain the Average Label-aware Margin (ALM) as follows: $\text{ALM}^{(t)}(\mathbf{x}, y) = \frac{1}{t} \sum_{r=1}^{t} \text{LM}^{(r)}(\mathbf{x}, y)$.

**Mitigating the harmful effect of mislabeled examples** We propose a weighted cross entropy loss during training and assign higher weights for high-ALM samples and lower weights otherwise. Let $N^t = \{\mathbf{x}_i \mid \text{ALM}^{(t)}(\mathbf{x}_i, y_i) < 0\}$ be the set of samples at iteration $t$ that have negative ALMs and $\text{ALM}(N^t)$ be the distribution of their ALMs. At $t$, we scale down the loss on samples from $N^t$ whose ALM is below the mean of the ALM distribution. We assume that samples with ALM above the mean are hard but clean and we do not reduce their importance (same as for positive ALM). Specifically, we dynamically fit a truncated Gaussian distribution of mean $\mu_t$ and variance $\sigma_t$ at iteration $t$ on all samples with ALM under the mean and assign a weight for each sample $\mathbf{x}_i$ as follows:

$$\lambda_{CE}^t(\mathbf{x}_i, y_i) = \begin{cases} \exp\left(-\frac{(\text{ALM}^{(t)}(\mathbf{x}_i, y_i) - \mu_t)^2}{2\sigma_t^2}\right) \text{ if } \mathbf{x}_i \in N^t \\ \qquad \text{and } \text{ALM}^t(\mathbf{x}_i, y_i) < \mu_t \\ 1 \quad \text{otherwise} \end{cases} \tag{4}$$

During training, we estimate the mean $\mu_t$ and variance $\sigma_t$ using the model's historical predictions:

$$\mu_t = \frac{1}{|N^t|} \sum_{(\mathbf{x}_i, y_i) \in N^t} \text{ALM}^{(t)}(\mathbf{x}_i, y_i) \tag{5}$$

$$\sigma_t = \frac{1}{|N^t|} \sum_{(\mathbf{x}_i, y_i) \in N^t} (\text{ALM}^{(t)}(\mathbf{x}_i, y_i) - \mu_t)^2 \tag{6}$$

Intuitively, a low weight for an example indicates that the example produced an ALM that is consistently below the mean of the negative ALM distribution. As we have shown, such examples are potentially mislabeled and may hurt generalization. Thus, at each training iteration $t$ we simply rescale the cross entropy loss, assigning lower weight to potentially mislabeled examples:

$$\mathcal{L}_{wCE} = \sum_{i=1}^{|B|} \lambda_{CE}^t(\mathbf{x}_i, y_i) \cdot H(\theta(\mathbf{x}_i), y_i) \tag{7}$$

where $\theta(\mathbf{x}_i)$ is the class distribution of model $\theta$ on example $\mathbf{x}_i$, $|B|$ is the batch size, and $H$ is the cross-entropy. To better distinguish between easily confusable classes, we extend the supervised contrastive loss by Gunel et al. (2020) and propose a label-aware supervised contrastive loss:

$$\mathcal{L}_{LSCL} = \sum_{i=1}^{|B|} H(\Pi(\mathbf{x}_i), y_i) + \sum_{i=1}^{|B|} \frac{-1}{|P_{\mathbf{x}_i}|} \sum_{p \in P_{\mathbf{x}_i}} \log \frac{w_{\mathbf{x}_i, y_{\mathbf{x}_i}} \cdot \exp(h_{\mathbf{x}_i}^\theta \cdot h_p^\theta)}{\sum_{s \in B; y_s \neq y_{\mathbf{x}_i}} w_{\mathbf{x}_i, y_s} \cdot \exp(h_{\mathbf{x}_i}^\theta \cdot h_s^\theta)} \tag{8}$$

where $B$ is the current batch, $P_{\mathbf{x}_i}$ is the set of positives $p$ for example $\mathbf{x}_i$ (i.e., in the context of supervised contrastive learning the positives are all examples that belong to the same class as $\mathbf{x}_i$ and its augmentations (Khosla et al., 2020)). $h_{\mathbf{x}_i}^\theta$ is the embedding of $\mathbf{x}_i$ produced by our classifier $\theta$. $w_{\mathbf{x}_i, y_{\mathbf{x}_i}}$ and $w_{\mathbf{x}_i, y_s}$ represent the soft-assignment of sample $\mathbf{x}_i$ to its assigned label $y_{\mathbf{x}_i}$ and to the non-assigned label $y_s$ where $y_s \neq y_{\mathbf{x}_i}$. To obtain these soft-assignments we simply utilize a projection layer on top of $\Pi$ followed by softmax, which produces the weights $w_{\mathbf{x}_i, y_s}$.

The final loss in LANE is the sum of the weighted cross entropy loss and the contrastive loss:

$$\mathcal{L} = \mathcal{L}_{wCE} + \mathcal{L}_{LSCL} \tag{9}$$

Appendix A shows the architecture of LANE.

## 4 EXPERIMENTS

### 4.1 DATASETS

We use the following datasets in our experiments: **1. Empathetic Dialogues (Empath)** (Rashkin et al., 2019), **2. GoEmotions (GoEmo)** (Demszky et al., 2020), **3. ISEAR** (Scherer & Wallbott, 1994), **4. CancerEMO (CEmo)** (Sosea & Caragea, 2020), **5. RCV1** (Lewis et al., 2004), **6. Sci-HTC** (Sadat & Caragea, 2022), **7. SST-5** (Socher et al., 2013b), **8. Amazon Review (Amazon R)** (McAuley & Leskovec, 2013), **9. Yelp Review (Yelp)** (Asghar, 2016), **10. Yahoo Answer (Yahoo)** (Chang et al., 2008). We provide details of these datasets in Appendix B.

### 4.2 EXPERIMENTAL SETUP

We evaluate the effectiveness of LANE on the above datasets under two label noise setups: **1)** Original datasets, where the label noise comes from annotation errors in the dataset collection process; and **2)** 20% noise, where we randomly shuffle the labels of 20% of the training data (an additional setup of 40% random noise is shown in Appendix C). We use the HuggingFace Transformers (Wolf et al., 2020) library for our BERT implementation. Both $\theta$ and $\Pi$ are BERT base uncased models. The datasets we consider make their train/validation/test splits available, hence, we use the provided splits in our experiments. Similar to Khosla et al. (2020), to expand the positive set of examples in the contrastive loss, we augment our data using synonym replacement (Kolomiyets et al., 2011), SwitchOut (Wang et al., 2018), and backtranslation (Tiedemann & Thottingal, 2020). For all datasets we follow the evaluation metrics used in the works introducing the datasets. The initial batch size is set to 32, hence the total batch size (i.e., including augmentations) is 256. In

| Dataset | Empath | GoEmo | ISEAR | CEmo | RCV1 |
|---------|--------|-------|-------|------|------|
| BASE | $58.5 \pm 1.2$ | $63.6 \pm 1.2$ | $71.5 \pm 0.6$ | $75.8 \pm 0.8$ | $56.8 \pm 0.8$ |
| E2L | $57.6 \pm 0.8$ | $63.2 \pm 1.2$ | $71.3 \pm 0.7$ | $75.9 \pm 0.9$ | $54.3 \pm 1.1$ |
| H2L | $58.9 \pm 1.4$ | $64.2 \pm 0.7$ | $72.0 \pm 0.6$ | $76.3 \pm 1.3$ | $55.8 \pm 1.4$ |
| AMG | $59.0 \pm 0.6$ | $\underline{64.8 \pm 0.6}$ | $73.4 \pm 0.5$ | $76.1 \pm 0.8$ | $52.3 \pm 1.1$ |
| NSE | $58.1 \pm 1.9$ | $63.8 \pm 1.1$ | $72.2 \pm 0.8$ | $76.2 \pm 0.7$ | $55.7 \pm 1.3$ |
| PLF | $58.4 \pm 1.1$ | $63.4 \pm 0.8$ | $71.9 \pm 1.2$ | $75.9 \pm 0.6$ | $56.7 \pm 2.2$ |
| AUM | $58.4 \pm 0.6$ | $63.1 \pm 1.3$ | $71.8 \pm 0.8$ | $76.0 \pm 0.9$ | $56.3 \pm 0.6$ |
| LCL | $59.1 \pm 1.0$ | $\underline{64.8 \pm 0.7}$ | $72.4 \pm 0.5$ | $76.5 \pm 0.9$ | $\underline{57.9 \pm 0.6}$ |
| SCL | $58.9 \pm 0.7$ | $62.8 \pm 1.1$ | $71.5 \pm 0.9$ | $76.2 \pm 0.6$ | $56.9 \pm 1.7$ |
| DISC | $\underline{59.4 \pm 0.9}$ | $63.2 \pm 1.4$ | $72.3 \pm 1.3$ | $76.4 \pm 1.1$ | $56.5 \pm 1.4$ |
| UNICON | $58.4 \pm 0.7$ | $63.1 \pm 0.9$ | $72.5 \pm 1.1$ | $76.6 \pm 1.3$ | $56.9 \pm 1.1$ |
| HMW | $57.6 \pm 1.1$ | $62.8 \pm 1.6$ | $70.4 \pm 1.4$ | $\underline{77.1 \pm 1.3}$ | $56.7 \pm 1.5$ |
| LANE | $\mathbf{60.8 \pm 0.9}$ | $\mathbf{66.5 \pm 0.5}$ | $\mathbf{74.3 \pm 0.4}$ | $\mathbf{78.2 \pm 0.7}$ | $\mathbf{59.3 \pm 0.9}$ |
| DATASET | SciHTC | SST-5 | Amazon R | Yelp | Yahoo |
| BASE | $32.5 \pm 1.75$ | $56.3 \pm 0.6$ | $67.5 \pm 0.6$ | $65.9 \pm 0.6$ | $75.4 \pm 0.6$ |
| E2L | $31.6 \pm 1.5$ | $55.7 \pm 1.1$ | $62.9 \pm 0.9$ | $62.8 \pm 2.3$ | $70.4 \pm 1.5$ |
| H2L | $32.2 \pm 1.1$ | $56.6 \pm 1.4$ | $67.9 \pm 0.8$ | $62.3 \pm 1.7$ | $74.1 \pm 1.8$ |
| AMG | $30.6 \pm 1.1$ | $55.1 \pm 1.3$ | $67.4 \pm 1.1$ | $65.1 \pm 1.5$ | $72.3 \pm 1.7$ |
| NSE | $32.8 \pm 1.5$ | $54.1 \pm 1.1$ | $65.8 \pm 1.7$ | $65.1 \pm 1.3$ | $74.6 \pm 1.1$ |
| PLF | $32.2 \pm 1.4$ | $55.7 \pm 1.1$ | $67.4 \pm 2.1$ | $65.8 \pm 1.8$ | $74.8 \pm 1.6$ |
| AUM | $31.2 \pm 2.63$ | $56.4 \pm 0.9$ | $66.4 \pm 0.6$ | $\underline{68.1 \pm 0.6}$ | $72.9 \pm 0.6$ |
| LCL | $\underline{33.1 \pm 1.42}$ | $\underline{57.6 \pm 0.9}$ | $68.2 \pm 0.6$ | $66.8 \pm 0.6$ | $76.8 \pm 0.6$ |
| SCL | $32.7 \pm 1.1$ | $56.8 \pm 1.5$ | $67.8 \pm 1.3$ | $66.1 \pm 1.7$ | $75.3 \pm 1.1$ |
| DISC | $32.8 \pm 1.5$ | $56.7 \pm 1.3$ | $67.8 \pm 2.4$ | $66.4 \pm 2.2$ | $75.1 \pm 1.7$ |
| UNICON | $32.7 \pm 1.1$ | $56.5 \pm 1.6$ | $67.5 \pm 1.4$ | $67.9 \pm 1.3$ | $77.1 \pm 1.5$ |
| HMW | $31.6 \pm 1.4$ | $57.2 \pm 1.1$ | $67.4 \pm 2.2$ | $68.1 \pm 1.7$ | $\underline{77.3 \pm 1.8}$ |
| LANE | $\mathbf{34.1 \pm 0.87}$ | $\mathbf{58.9 \pm 0.4}$ | $\mathbf{69.7 \pm 0.6}$ | $\mathbf{69.2 \pm 0.6}$ | $\mathbf{78.4 \pm 0.6}$ |

Table 2: Results of LANE on the original datasets. The results are averaged across five runs and standard deviations are provided. Best results are shown in **bold blue** and second best are underlined.

terms of computational overhead, the auxiliary network in LANE increases training compute by approximately $1.8\times$ compared to a standard BERT fine-tuning run. Despite this, the model maintains a similar convergence rate in terms of epochs and remains compatible with a single consumer GPU. This positions LANE as a computationally efficient alternative to the resource-intensive process of utilizing Large Language Models for noisy label correction. In our training setup, we only scale down the importance of examples during training if their ALM is below a threshold that we set as the ALM mean of examples with negative ALMs (Eq. 5). We also experimented with different ALM thresholds such as $0$, but observed worse performance than using the mean (see Appendix D).

## 4.3 BASELINE MODELS

We use BERT (Devlin et al., 2019) base uncased model in all experiments (denoted by BASE). We compare LANE against methods that use training dynamics to assess the data quality, as well as approaches focused on exploiting the relationships between classes and approaches aimed at learning under label noise: **Data Cartography** (E2L, H2L, AMG) (Swayamdipta et al., 2020), **Noise Layer** (NSE) (Goldberger & Ben-Reuven, 2016), **Peer Loss Function (PLF)** (Liu & Guo, 2020), **Area Under the Margin** (AUM) (Pleiss et al., 2020), **Supervised Contrastive Learning** (SCL) Gunel et al. (2020), **Label-aware Contrastive Learning** (LCL) Suresh & Ong (2021), **DISC** (Li et al., 2023), **UNICON** (Karim et al., 2022), and **Hyperspherical Margin Weighting** (HMW) (Zhang et al., 2024). We provide more details into these baselines in Appendix E.1.

## 5 RESULTS

**Results on Original Datasets** We show the results on the original datasets in Table 2. We make the following observations. **LANE outperforms the baselines in all setups**. We observe improvements of $1.1\%$ weighted F1 on CancerEmo, $1.4\%$ weighted F1 on RCV1, $1.5\%$ accuracy on Amazon Review and $1.1\%$ accuracy on Yahoo over the best performing baseline. Notably, over the base BERT model, we see a $2.9\%$ weighted F1 improvement on GoEmotions and $3.0\%$ improvement on Yahoo. We note that LCL, which leverages inter-class relations through the label-aware contrastive learning loss is the best performing baseline in 5 out of the 10 datasets. Since LANE utilizes similar

| Dataset | Empath | GoEmo | ISEAR | CEmo | RCV1 |
|---|---|---|---|---|---|
| BASE | $11.6 \pm 3.4$ | $21.5 \pm 2.8$ | $37.6 \pm 3.0$ | $46.7 \pm 1.9$ | $44.4 \pm 3.8$ |
| E2L | $10.3 \pm 0.8$ | $22.6 \pm 1.2$ | $37.1 \pm 0.7$ | $47.5 \pm 0.9$ | $44.3 \pm 1.5$ |
| H2L | $10.6 \pm 1.4$ | $21.8 \pm 0.7$ | $37.3 \pm 0.6$ | $47.9 \pm 1.3$ | $45.8 \pm 2.4$ |
| AMG | $11.4 \pm 1.2$ | $22.1 \pm 0.6$ | $36.9 \pm 0.5$ | $48.4 \pm 0.8$ | $45.9 \pm 2.7$ |
| NSE | $10.2 \pm 1.9$ | $15.6 \pm 1.1$ | $36.4 \pm 0.8$ | $44.2 \pm 0.7$ | $44.9 \pm 1.8$ |
| PLF | $12.5 \pm 1.7$ | $16.7 \pm 1.3$ | $36.8 \pm 1.3$ | $50.1 \pm 0.9$ | $48.2 \pm 1.7$ |
| AUM | $\underline{14.5 \pm 0.6}$ | $\underline{23.5 \pm 1.3}$ | $38.6 \pm 0.8$ | $49.8 \pm 0.9$ | $47.6 \pm 2.7$ |
| SCL | $10.4 \pm 1.4$ | $21.4 \pm 1.3$ | $37.3 \pm 0.9$ | $46.4 \pm 1.1$ | $45.2 \pm 1.5$ |
| LCL | $10.8 \pm 3.24$ | $22.1 \pm 5.1$ | $38.3 \pm 1.5$ | $46.6 \pm 1.2$ | $47.2 \pm 2.2$ |
| DISC | $11.3 \pm 1.0$ | $22.5 \pm 0.7$ | $\mathbf{40.5 \pm 0.5}$ | $50.3 \pm 0.9$ | $47.1 \pm 2.2$ |
| UNICON | $10.4 \pm 1.4$ | $21.9 \pm 1.2$ | $39.5 \pm 0.9$ | $42.3 \pm 0.9$ | $\underline{49.2 \pm 2.3}$ |
| HMW | $12.4 \pm 1.9$ | $22.0 \pm 1.5$ | $38.1 \pm 2.1$ | $\underline{50.7 \pm 2.2}$ | $48.2 \pm 1.8$ |
| LANE | $\mathbf{15.9 \pm 1.3}$ | $\mathbf{24.3 \pm 1.2}$ | $\underline{40.4 \pm 0.8}$ | $\mathbf{52.5 \pm 0.9}$ | $\mathbf{49.4 \pm 2.1}$ |

| DATASET | SciHTC | SST-5 | Amazon R | Yelp | Yahoo |
|---|---|---|---|---|---|
| BASE | $24.5 \pm 4.6$ | $48.9 \pm 3.7$ | $61.5 \pm 1.5$ | $60.7 \pm 1.3$ | $64.8 \pm 1.7$ |
| E2L | $24.1 \pm 2.4$ | $48.2 \pm 2.7$ | $60.7 \pm 2.4$ | $62.3 \pm 2.9$ | $64.9 \pm 3.1$ |
| H2L | $26.7 \pm 2.3$ | $48.7 \pm 1.9$ | $60.9 \pm 2.3$ | $62.6 \pm 2.1$ | $65.7 \pm 1.8$ |
| AMG | $26.9 \pm 1.4$ | $49.4 \pm 1.5$ | $61.3 \pm 2.4$ | $62.9 \pm 2.3$ | $66.5 \pm 1.8$ |
| NSE | $26.7 \pm 4.3$ | $50.4 \pm 4.1$ | $61.7 \pm 3.5$ | $63.5 \pm 3.3$ | $67.2 \pm 2.5$ |
| PLF | $\underline{29.7 \pm 1.7}$ | $51.9 \pm 2.8$ | $62.1 \pm 2.3$ | $\underline{64.7 \pm 2.7}$ | $\underline{68.1 \pm 2.3}$ |
| AUM | $27.4 \pm 4.2$ | $50.4 \pm 2.5$ | $\underline{62.4 \pm 1.7}$ | $63.3 \pm 1.4$ | $65.9 \pm 2.4$ |
| LCL | $24.2 \pm 3.9$ | $48.5 \pm 5.7$ | $61.7 \pm 2.4$ | $63.1 \pm 3.1$ | $65.9 \pm 3.0$ |
| SCL | $24.1 \pm 3.4$ | $51.5 \pm 3.2$ | $62.3 \pm 3.5$ | $63.7 \pm 3.9$ | $66.8 \pm 2.5$ |
| DISC | $27.5 \pm 2.1$ | $\underline{51.7 \pm 2.6}$ | $62.1 \pm 2.7$ | $63.2 \pm 2.5$ | $67.3 \pm 2.1$ |
| UNICON | $28.9 \pm 3.4$ | $50.8 \pm 3.1$ | $61.5 \pm 3.7$ | $62.3 \pm 3.9$ | $64.2 \pm 3.7$ |
| HMW | $28.7 \pm 1.5$ | $51.3 \pm 1.8$ | $61.2 \pm 1.1$ | $62.5 \pm 1.9$ | $66.3 \pm 2.2$ |
| LANE | $\mathbf{30.5 \pm 2.97}$ | $\mathbf{53.1 \pm 1.6}$ | $\mathbf{63.1 \pm 2.3}$ | $\mathbf{65.2 \pm 3.1}$ | $\mathbf{68.9 \pm 2.5}$ |

Table 3: Performance of LANE in the $20\%$ noise setting. The reported results are averaged across five runs and standard deviations are provided. Best results are shown in **bold blue** and second best are underlined.

inter-class relations during training, we postulate improvements over LCL arise from correctly identifying mislabeled or ambiguous examples and eliminating their harmful effect during training.

**Results on 20% Noise Datasets** The results obtained on the $20\%$ noise (20N) datasets where $20\%$ of the labels are intentionally flipped are shown in Table 3. We observe that this setup is significantly more challenging for the model. For instance, on Empathetic Dialogues, the weighted F1 of the BASE model drops from $58.5\%$ on the original dataset to $11.6\%$ on the 20N dataset, with a similar trend on all the other datasets. However, even in this more challenging setup, LANE still outperforms the majority of baselines in all setups. For example, on SST-5, LANE outperforms AUM in accuracy by $2.7\%$, DISC by $1.4\%$, UNICON by $2.3\%$, and SCL by $1.6\%$. The improvements over the base model are larger, with an average performance increase of $4.11\%$.

# 6 ANALYSIS

**Ablation Study** For our ablation, we design a version of LANE that uses averaged margins instead of ALMs so that the semantic relations are not incorporated into the model. We achieve this by replacing the ALM term with AUM in Equations 4, 5, and 6 and denote this method by LANE$^{-sim}$. This approach utilizes all training samples with per-sample AUM weight. Second, we design a version of LANE that does not use weighted cross entropy in Equation 7, i.e., $\lambda_{CE}^{t} = 1$. We denote this method by LANE$^{-alm}$. Third, we compare LANE against the vanilla AUM approach, which removes examples from the training set that have low AUMs.

We show the results on both the original and $20\%$ noise (20N) datasets in Table 4. We observe that LANE consistently outperforms LANE$^{-sim}$, LANE$^{-alm}$, and AUM in all settings. The improvements are particularly noticeable in the more challenging 20N setup. For instance, on the RCV1 dataset, which has a large number of classes, LANE improves the micro F1 score to $49.4\%$, a boost of $4.2\%$ over LANE$^{-sim}$, $3.2\%$ over LANE$^{-alm}$, and $1.8\%$ over AUM. The trend also holds on the original datasets. On SciHTC, LANE achieves an F1 score of $34.1\%$, outperforming AUM by $2.9\%$ and LANE$^{-sim}$ by $1.7\%$. These results show that our proposed Average Label-aware Margin and the semantics-aware contrastive loss play an important role in the success of LANE.

| DATASET: | Original Dataset | | | | 20% Noise | | | |
|---|---|---|---|---|---|---|---|---|
| | **Empath** | **SciHTC** | **Amazon R** | **RCV1** | **Empath** | **SciHTC** | **Amazon R** | **RCV1** |
| LANE$^{-sim}$ | $58.7_{1.1}$ | $\underline{32.4_{0.8}}$ | $66.8_{0.8}$ | $57.3_{0.8}$ | $\underline{14.7_{1.1}}$ | $28.5_{0.8}$ | $61.2_{0.8}$ | $45.2_{0.8}$ |
| LANE$^{-alm}$ | $\underline{59.1_{0.9}}$ | $32.1_{1.2}$ | $\underline{68.2_{1.2}}$ | $\underline{57.9_{1.4}}$ | $13.8_{0.9}$ | $\underline{29.3_{1.2}}$ | $61.3_{1.2}$ | $46.2_{1.4}$ |
| AUM | $58.2_{0.7}$ | $31.2_{3.7}$ | $66.1_{1.4}$ | $56.3_{2.2}$ | $14.5_{0.6}$ | $27.4_{4.2}$ | $\underline{62.4_{1.7}}$ | $\underline{47.6_{2.7}}$ |
| LANE | $\mathbf{60.8_{1.5}}$ | $\mathbf{34.1_{2.31}}$ | $\mathbf{69.7_{2.1}}$ | $\mathbf{59.3_{2.3}}$ | $\mathbf{15.9_{1.5}}$ | $\mathbf{30.5_{2.41}}$ | $\mathbf{63.1_{2.1}}$ | $\mathbf{49.4_{1.8}}$ |

Table 4: Ablation study: comparison between LANE, LANE$^{-sim}$, LANE$^{-alm}$ and vanilla AUM on the datasets using original data and $20\%$ noise. Best results are shown in **bold blue** and second best are underlined. Subscripts denote standard deviation.

| DATASET: | Amazon R | Yelp | AG News | Yahoo |
|---|---|---|---|---|
| DIVIDEMIX | $60.66 \pm 3.1$ | $64.66 \pm 2.5$ | $89.05 \pm 1.1$ | $69.17 \pm 1.1$ |
| CO-TEACHING | $60.62 \pm 3.7$ | $63.65 \pm 2.7$ | $89.04 \pm 1.6$ | $68.69 \pm 3.7$ |
| LANE | $\mathbf{69.7 \pm 1.3}$ | $\mathbf{69.2 \pm 1.2}$ | $\mathbf{90.01 \pm 0.8}$ | $\mathbf{78.4 \pm 0.8}$ |

Table 5: Performance of LANE compared to Dual Network Approaches.

**Comparison with 2-Network Approaches** To further contextualize LANE's performance, we compare it against other popular two-network architectures designed to handle label noise: Co-teaching Han et al. (2018) and DivideMix Li et al. (2020), presented in detail in Appendix E.2. We evaluated these approaches against LANE on four benchmark datasets, with the results detailed in Table 5. The analysis reveals that LANE consistently and significantly outperforms both Co-teaching and DivideMix across all tested datasets. On the sentiment classification tasks, LANE achieves an accuracy of 69.7% on Amazon Review and 69.2% on Yelp Review, surpassing the next-best method, DivideMix, by substantial margins of over 9 and 4.5 percentage points, respectively. This performance gap is even more pronounced on the Yahoo dataset, where LANE's 78.4% accuracy represents a greater than 9-point improvement over DivideMix. Even on the AG News dataset, where all models perform well, LANE still establishes a new state-of-the-art result with 90.01% accuracy. These results strongly suggest that LANE's use of label-aware margins for dynamic sample weighting is a very effective strategy for mitigating label noise.

**Understanding LANE's Weight Distribution** A core hypothesis behind LANE is its ability to differentiate between clean and noisy labels by assigning lower weights to samples it perceives as incorrectly labeled. To validate this, we analyze the distribution of weights assigned by LANE to both correctly and incorrectly labeled examples. This experiment is conducted on the RCV1, Amazon Review, and SciHTC datasets, each injected with $20\%$ label noise. Figure 1 illustrates these distributions. The results provide clear and compelling evidence supporting our hypothesis.

Across all three datasets, the weight distribution for incorrectly labeled examples (shown in red) is heavily skewed towards the left, with a distinct peak around weight 0. For instance, in the RCV1 dataset, over $40\%$ of all incorrect samples are assigned a weight in the $[0, 0.1]$ interval. This demonstrates that LANE is highly effective at identifying noisy samples and drastically reducing their impact on the model's training process by assigning them near-zero weights. Conversely, the weight distribution for correctly labeled examples (shown in blue) is skewed towards the right, with the majority of weights concentrated in the higher ranges (approximately 0.4 to 0.9). This indicates

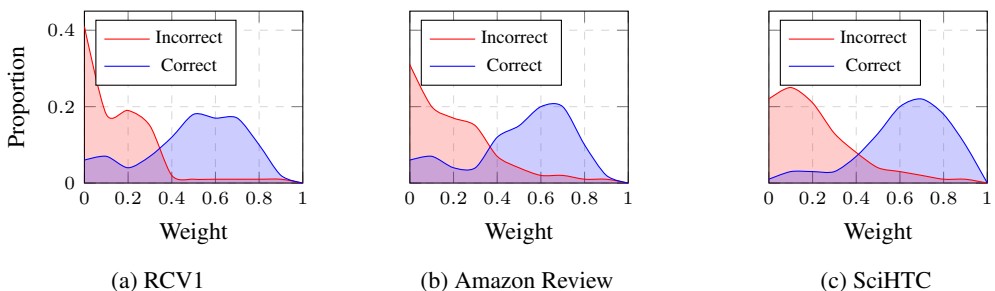

(a) RCV1  (b) Amazon Review  (c) SciHTC

Figure 1: Distribution of Correct and Incorrect Examples by Weight

| Method | Empath | GoEmo | ISEAR | CEmo | RCV1 | SciHTC | SST-5 | Amazon | Yelp | Yahoo |
|---|---|---|---|---|---|---|---|---|---|---|
| BASE (BERT) | $58.5 \pm 1.2$ | $63.6 \pm 1.2$ | $71.5 \pm 0.6$ | $75.8 \pm 0.8$ | $56.8 \pm 0.8$ | $32.5 \pm 1.7$ | $56.3 \pm 0.6$ | $67.5 \pm 0.6$ | $65.9 \pm 0.6$ | $75.4 \pm 0.6$ |
| LANE (BERT) | $60.8 \pm 0.9$ | $66.5 \pm 0.5$ | $74.3 \pm 0.4$ | $78.2 \pm 0.7$ | $59.3 \pm 0.9$ | $34.1 \pm 0.8$ | $58.9 \pm 0.4$ | $69.7 \pm 0.6$ | $69.2 \pm 0.6$ | $78.4 \pm 0.6$ |
| BASE (RoBERTa) | $59.1 \pm 1.1$ | $64.2 \pm 1.4$ | $71.4 \pm 0.8$ | $76.2 \pm 0.7$ | $57.3 \pm 0.6$ | $32.1 \pm 1.9$ | $56.5 \pm 0.8$ | $67.9 \pm 0.5$ | $66.1 \pm 0.7$ | $75.7 \pm 0.9$ |
| LANE (RoBERTa) | $61.2 \pm 0.7$ | $66.8 \pm 1.2$ | $74.5 \pm 0.9$ | $78.3 \pm 2.1$ | $59.1 \pm 1.3$ | $34.6 \pm 0.9$ | $59.0 \pm 0.6$ | $70.1 \pm 0.7$ | $69.8 \pm 0.5$ | $78.9 \pm 0.7$ |

Table 6: Performance of LANE using different backbone architectures (BERT vs. RoBERTa).

that the model preserves the valuable signal from clean data by assigning these samples high importance. Therefore, the clear separation between the two distributions validates the core mechanism of LANE. The label-aware margin effectively serves as a reliable proxy for label correctness, allowing the model to dynamically filter out noise and prioritize learning from clean, high-quality examples.

Note that while Figure 1 illustrates the continuous weight distribution across these samples, LANE still enforces the ALM thresholding mechanism during actual training (Equation 4); therefore, samples with an ALM above the dynamic threshold are not penalized and simply retain a full weight of 1.

**Applicability Beyond BERT** In our primary experiments, we selected BERT Devlin et al. (2019) as the backbone architecture to ensure a fair and direct comparison with an extensive suite of established baselines (e.g., AUM, DivideMix, UNICON), which predominantly utilize BERT as the standard in the noise learning literature. However, LANE is fundamentally model-agnostic. The core mechanism, the Label-Aware Margin (LM), relies exclusively on the logits and latent embeddings produced by the network. Because these are standard outputs for any deep learning classifier, LANE can be seamlessly integrated with other architectures, including RoBERTa Liu et al. (2019), DeBERTa He et al. (2020), or even vision models like ResNet He et al. (2016). To empirically validate this generalizability, we conducted supplementary experiments replacing the BERT backbone with RoBERTa. As shown in Table 6, LANE integrated with RoBERTa performs robustly, matching or exceeding the performance gains observed with BERT. This confirms that LANE's dynamic sample weighting is an architectural-independent strategy for mitigating label noise.

**Additional Analyses** We conduct several supplementary evaluations in the Appendix to further validate the robustness and adaptability of LANE across various scenarios. **Performance Under Extreme Noise (40%):** In Appendix C, we evaluate LANE under a severe noise setting where 40% of the training labels are randomly shuffled, demonstrating that LANE maintains its effectiveness and significantly outperforms baselines like AUM and DISC. **ALM Threshold Analysis:** We analyze the impact of dynamically learning the Gaussian re-weighting parameters versus using fixed ALM thresholds ($\mu = 0$ and $\mu = -1$, with $\sigma = 1$) in Appendix D, confirming that dynamically inferring the threshold yields the best performance. **Performance Against LLMs:** We compare LANE's noise robustness against few-shot Large Language Models (ChatGPT and Llama-2 13B) under a 20% label noise setup in Appendix F, where LANE consistently outperforms the evaluated LLMs across the majority of datasets. **Semantic-Aware Label Noise:** To simulate realistic human annotation errors, we evaluate LANE on datasets where noise is injected based on inter-class semantic similarities rather than uniform randomness in Appendix G, showing that LANE effectively mitigates this ambiguity-driven label noise better than existing baselines. **Noise Correction Comparison:** Finally, we benchmark LANE against a recent state-of-the-art interpolation-based noise correction approach by Xu et al. (2025) in Appendix H, highlighting that LANE outperforms this recent method on four out of the five evaluated datasets.

## 7 CONCLUSION

In this work, we introduced LANE, a new approach that boosts the capabilities of deep learning models when learning under increased label noise. LANE leverages the inter-class semantic similarities and utilizes training dynamics to boost the performance in fine-grained text classification. We tested LANE on ten fine-grained text classification datasets where it obtained improvements in performance over strong baselines and prior works. In the future, we plan to extend our approach to other domains and data types, e.g., image classification and the legal domain. We make our code available to further research in this area.

**Acknowledgement** This research is supported in part by the NSF IIS award 2107518. Any opinions, findings, and conclusions expressed here are those of the authors and do not necessarily reflect the views of NSF. We thank our anonymous reviewers for their constructive feedback, which helped improve the quality of our paper.

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

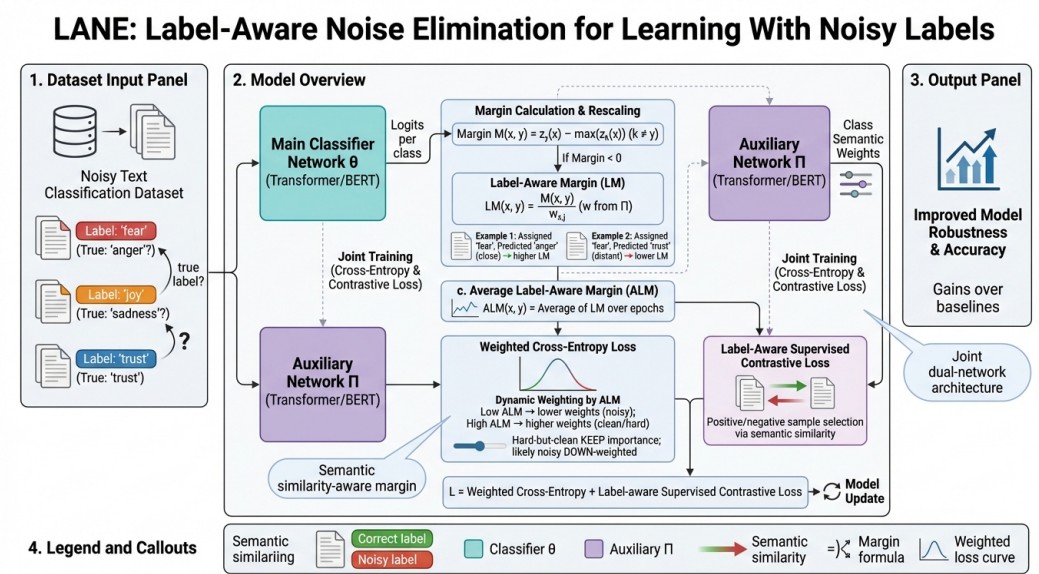

Figure 2: Architecture of LANE.

## A ARCHITECTURE

Figure 2 shows the architecture of our LANE approach.

## B DATASETS

We evaluate LANE on: **1. Empathetic Dialogues** (Rashkin et al., 2019), a dataset composed of conversations between a speaker and a listener annotated with 32 emotions. We consider solely the first turn of the conversation in our experiments, resulting in $22,000$ total examples. **2. GoEmotions** (Demszky et al., 2020), a sentence-level dataset created using Reddit comments and contains more than $58,000$ sentences annotated with 27 emotions. **3. ISEAR** (International Survey on Emotion Antecedents and Reactions) (Scherer & Wallbott, 1994), a dataset of $7,700$ personal experiences annotated with 7 emotions. **4. CancerEMO** (Sosea & Caragea, 2020), a dataset of $8,500$ examples collected from a cancer forum annotated at sentence level with the 8 basic Plutchik-8 (Plutchik, 1980) emotions. **5. RCV1** (Lewis et al., 2004), a large scale dataset composed of news stories labeled with a total of $105$ different topics. **6. SciHTC** (Sadat & Caragea, 2022), a dataset from $186,160$ scientific papers, annotated with $80$ topics. **7. SST-5** (Socher et al., 2013b), a dataset composed of $11,855$ sentences from movie reviews, annotated with five sentiment labels: *negative*, *somewhat negative*, *neutral*, *somewhat positive*, and *positive*. **8. Amazon Review** (McAuley & Leskovec, 2013), a sentiment classification dataset composed of $600,000$ training and $130,000$ test Amazon reviews annotated with 5 sentiment classes. **9. Yelp Review** (Asghar, 2016), a sentiment classification dataset with $130,000$ training and $10,000$ test samples annotated with the same 5 classes, and **10. Yahoo Answer** (Chang et al., 2008), a topic classification dataset with 10 topic classes, composed of $140,000$ training and $6,000$ test samples.

## C DATASETS WITH $40\%$ LABEL NOISE

We show in Table 7 results on the $40\%$ noise (40N) datasets. Results marked with $-$ indicate that the model did not convege. We notice that LANE stays effective across the ten datasets, and we observe that AUM yields poor results on this dataset with very high amounts of noise, indicating that it may not work in high-noise setups. For example, AUM outperforms DISC by an average of $1.5\%$ on 20N across the datasets whereas DISC outperforms AUM on 40N by a significant $2.9\%$. Critically, LANE outperforms both DISC and AUM on 40N by an average of $2.8\%$ and $4.75\%$, respectively.

| Dataset | Empathetic Dialogues (wF1) | GoEmotions (wF1) | ISEAR (wF1) | CancerEmo (wF1) | RCV1 (wF1) |
|---|---|---|---|---|---|
| BASE | – | – | – | – | – |
| E2L | – | – | – | – | – |
| H2L | – | – | – | – | – |
| AMG | – | – | – | – | – |
| NSE | – | – | – | – | $31.4 \pm 1.7$ |
| PLF | – | – | – | – | – |
| AUM | $10.4 \pm 0.6$ | $17.5 \pm 1.3$ | $27.8 \pm 0.8$ | $41.8 \pm 0.9$ | $32.5 \pm 1.3$ |
| LCL | – | – | – | – | – |
| SCL | – | – | – | – | – |
| DISC | $14.1 \pm 1.7$ | $19.6 \pm 0.7$ | $31.4 \pm 0.5$ | $47.6 \pm 0.9$ | $33.7 \pm 1.5$ |
| UNICON | $13.7 \pm 1.4$ | $17.4 \pm 1.2$ | $33.1 \pm 0.9$ | $46.5 \pm 0.9$ | $34.6 \pm 1.5$ |
| HMW | $11.2 \pm 3.5$ | $18.6 \pm 3.2$ | $27.6 \pm 3.7$ | $41.5 \pm 2.6$ | $34.9 \pm 2.4$ |
| LANE | $\mathbf{14.6 \pm 1.2}$ | $\mathbf{20.5 \pm 0.9}$ | $\mathbf{35.1 \pm 0.7}$ | $\mathbf{50.1 \pm 0.6}$ | $\mathbf{38.2 \pm 1.7}$ |
| DATASET | SciHTC (MF1) | SST-5 (Acc) | Amazon Review (Acc) | Yelp (Acc) | Yahoo (Acc) |
| BASE | – | – | – | – | – |
| E2L | – | – | – | – | – |
| H2L | – | – | – | – | – |
| AMG | – | – | – | – | – |
| NSE | $14.8 \pm 1.5$ | $41.6 \pm 2.3$ | – | $44.7 \pm 2.6$ | – |
| PLF | – | – | – | – | – |
| AUM | $17.2 \pm 1.4$ | $42.6 \pm 1.5$ | $51.4 \pm 1.1$ | $52.6 \pm 1.8$ | $42.7 \pm 1.9$ |
| LCL | – | – | – | – | – |
| SCL | – | – | – | – | – |
| DISC | $18.5 \pm 2.3$ | $43.8 \pm 1.8$ | $52.9 \pm 1.9$ | $53.8 \pm 2.3$ | $44.7 \pm 2.1$ |
| UNICON | $19.6 \pm 1.5$ | $43.1 \pm 1.6$ | $55.2 \pm 1.3$ | $53.9 \pm 1.7$ | $44.7 \pm 2.1$ |
| HMW | $17.4 \pm 1.1$ | $43.1 \pm 1.7$ | $53.1 \pm 2.3$ | $53.7 \pm 1.7$ | $42.8 \pm 2.2$ |
| LANE | $\mathbf{20.5 \pm 1.5}$ | $\mathbf{45.7 \pm 1.3}$ | $\mathbf{56.8 \pm 2.2}$ | $\mathbf{56.2 \pm 2.3}$ | $\mathbf{46.3 \pm 2.5}$ |

Table 7: Performance of LANE on the the ten benchmark datasets under 40% label noise. The reported results are averaged across five runs and standard deviations are provided. Best results are shown in **bold blue** and second best are underlined. Results marked with − indicate that the model did not converge.

## D    ANALYSIS OF THE ALM THRESHOLD

In our main experiments, we automatically learned the Gaussian re-weighting parameters from the data, as described in the main paper. Specifically, during training, we estimate the mean $\mu_t$ and variance $\sigma_t$ using the model's historical predictions. We also experimented with using fixed, hard-coded values for the threshold, setting $\mu$ to 0 and -1, with $\sigma = 1$. When a fixed threshold is used, we re-weight a sample if its Average Label-aware Margin (ALM) is less than the threshold (e.g., $ALM < 0$); otherwise, the sample's weight remains 1, per our weighting function (Equation 4).

We present the results of these experiments on the Yelp and Yahoo datasets in Table 8. The results show that the best performance is achieved when the parameters are inferred dynamically from the data's historical predictions, validating the approach used in our paper.

| Approach | $\mu = 0$ | $\mu = -1$ | Inferred from Data |
|---|---|---|---|
| Yelp | 64.4 | 64.1 | **65.2** |
| Yahoo | 68.1 | 65.1 | **68.9** |

Table 8: Performance comparison (Accuracy) on the Yelp and Yahoo datasets using different ALM thresholds ($\mu$). The "Inferred from Data" column uses the dynamic method from Equations 5 and 6 in the main paper.

## E    BASELINES

### E.1    SINGLE NETWORK APPROACHES

**Data Cartography**    Following (Swayamdipta et al., 2020), we identify three types of training examples: easy-to-learn (E2L), hard-to-learn (H2L), and ambiguous (AMG) and analyze the importance of each type to the training process by removing the other two types.

**Noise Layer**    Following (Goldberger & Ben-Reuven, 2016), we introduce a noise layer to the BERT model which we train for correct label estimation. We denote this model by NSE in our experiments.

| DATASET: | Empathetic Dialogues (wF1) | GoEmotions (mF1) | ISEAR (ACC) | CancerEMO (mF1) | RCV1 (mF1) |
|---|---|---|---|---|---|
| CHATGPT | 12.8 ± 3.1 | 21.4 ± 2.5 | 37.3 ± 1.1 | 48.9 ± 1.9 | 42.9 ± 4.6 |
| LLAMA-2 | 10.9 ± 3.7 | 20.4 ± 2.7 | 35.4 ± 1.6 | 50.2 ± 1.7 | 39.7 ± 1.8 |
| LANE | **15.9 ± 1.3** | **24.3 ± 1.2** | **40.4 ± 0.8** | **52.5 ± 0.9** | **49.4 ± 2.1** |

| DATASET: | SciHTC (MF1) | SST-5 (Acc) | Amazon Review (Acc) | Yelp (Acc) | Yahoo (Acc) |
|---|---|---|---|---|---|
| CHATGPT | 28.3 ± 5.0 | 49.6 ± 0.6 | 62.6 ± 0.9 | 64.5 ± 0.9 | 64.9 ± 0.9 |
| LLAMA-2 | 15.1 ± 5.2 | **54.2 ± 0.4** | 61.3 ± 2.3 | 62.3 ± 1.4 | 61.1 ± 2.3 |
| LANE | **30.5 ± 2.97** | 53.1 ± 1.6 | **63.1 ± 2.3** | **65.2 ± 3.1** | **68.9 ± 2.5** |

Table 9: Performance of LANE compared with LLMs. Best results are shown in **bold blue** and second best are underlined.

**Peer Loss Function** We also compare our method against Peer Loss Function (PLF) (Liu & Guo, 2020), a method that alters the training loss function to account for label noise.

**Area Under the Margin** We consider the AUM method (Pleiss et al., 2020) as one of our baselines. This method computes Area Under the Margin metric for each training example and eliminates low-AUM examples that are potentially noisy, using a fixed threshhold for elimination.

**Contrastive Learning** We compare LANE to the label-aware supervised contrastive learning (LCL) method proposed by Suresh & Ong (2021) and the traditional supervised contrastive learning (SCL) (Khosla et al., 2020).

**DISC** (Li et al., 2023) proposes an instance-specific dynamic thresholding mechanism that blocks access to specific training examples based on the momentum of each instance's memorization strength. Additionally, DISC proposes to correct the labels of potentially noisy examples.

**UNICON** (Karim et al., 2022) leverages semi-supervised learning (SSL) to mitigate the harmful effects of noisy labels by considering the potentially noisy labeled data as unlabeled examples in an SSL algorithm. UNICON also proposes a new selection mechanism for these unlabeled examples during training.

**Hyperspherical Margin Weighting (HMW)** (Zhang et al., 2024) is a sample weighting strategy that improves learning with noisy labels by using a novel metric called the Integrated Area Margin (IAM). To better distinguish clean but hard-to-learn examples from mislabeled ones, the IAM metric is constructed by combining two distinct margin-based signals: the established AUM ranking Pleiss et al. (2020) and a newly proposed Top-K Under the Margin (TKUM) ranking.

## E.2 DUAL NETWORK APPROACHES

**Co-teaching** Han et al. (2018) take a peer-teaching approach and simultaneously trains two deep neural networks. In each mini-batch, each network identifies samples it believes have a small loss (and are therefore likely to be correctly labeled) and feeds these "clean" samples to its peer network for subsequent training. This cross-training helps the models avoid overfitting to noisy labels that one network might have memorized.

**DivideMix** Li et al. (2020) reframes learning with noisy labels as a semi-supervised learning problem. It also maintains two diverged networks and at the start of each epoch, it uses a Gaussian Mixture Model on the per-sample loss distributions to dynamically divide the training data into a labeled set of likely clean samples and an unlabeled set of likely noisy samples. Each network then trains on the dataset division provided by the other, enhancing robustness.

## F PERFORMANCE AGAINST LLMS

We test our approach against few-shot large language models: ChatGPT and Llama-2 13B (Touvron et al., 2023) to compare the robustess to label noise of LANE with that of popular LLMs in 20% noise setup and show results in Table 9. For all datasets except SciHTC we fit a large number of examples in the prompt and set the number of few-shot examples to 100. We use only 10 few-shot examples for SciHTC since the examples (i.e., paper abstracts) are much longer and exceed the context window. Similar to the original 20% noise setup, 20% of the few-shot examples are purposefully mislabeled. To account for the variance produced by the particular few-shot examples selected, we

run ChatGPT 10 times with different few-shot examples in the prompt and report average values. Similarly, we run Llama-2 20 times with different few-shot examples. We observe that LANE outperforms the LLMs on all datasets except SST-5. Notably, LANE improves upon Llama-2 by $15.4\%$ on SciHTC and by $18.7\%$ on RCV1 and improves the performance over ChatGPT by $3.1\%$ accuracy on ISEAR and $6.5\%$ micro F1 on RCV1. Among the LLMs, ChatGPT obtains the best results, outperforming Llama-2 especially in complex tasks such as RCV1 and SciHTC. Concretely, ChatGPT obtains $28.3\%$ macro F1 on SciHTC, a $13.2\%$ improvement over Llama-2.

## G    SEMANTIC-AWARE LABEL NOISE GENERATION DETAILS

In fine-grained classification, annotation errors are rarely uniformly random; human annotators are more likely to confuse semantically similar classes. To evaluate LANE under a more realistic ambiguity setting, we designed a *Semantic-Aware Noise* setup. Instead of flipping labels randomly, we biased the label-flipping probability so that instances are much more likely to be mislabeled as a semantically similar class rather than a completely unrelated one. The semantic noise datasets are generated as described below. We evaluated LANE against the best-performing baselines on datasets with well-defined class similarities. As shown in Table 10, LANE consistently outperforms all baselines in this challenging scenario, confirming that the Label-Aware Margin effectively mitigates realistic, ambiguity-driven label noise.

| Method | Empath | GoEmo | Yelp | CEmo | SST-5 |
|--------|--------|-------|------|------|-------|
| BASE | 50.3 | 57.2 | 64.2 | 73.4 | 53.3 |
| AMG | 52.7 | 60.3 | 66.2 | 74.8 | 54.1 |
| AUM | 51.2 | 61.1 | 66.5 | 74.1 | 52.7 |
| LCL | 53.6 | 60.1 | 65.4 | 73.9 | 53.2 |
| DISC | 52.4 | 60.2 | 64.3 | 74.2 | 53.6 |
| LANE | **54.2** | **61.4** | **67.4** | **75.1** | **54.3** |

Table 10: Performance comparison under Semantic-Aware Noise.

**Semantic-aware label noise generation:**    In standard noisy label benchmarks, synthetic noise is typically introduced by flipping a clean label to any other available class with uniform probability. However, to account for the realistic scenario of how human annotation errors occur (i.e., being heavily biased toward semantically similar classes), instead of a uniform distribution, we define the label-flipping probability using an exponential decay function parameterized by a semantic distance metric, $d(y, k)$, between the true label $y$ and a candidate noisy label $k$. The definition of the semantic distance $d(y, k)$ depends on the inherent structure of the dataset's label space:

*Sentiment Datasets (SST-5, Yelp):* For datasets with an ordinal scale (e.g., highly negative to highly positive), the distance is defined as the absolute difference between their discrete scalar values. For example, on a 5-point scale, the distance between a 1-star review ($y = 1$) and a 2-star review ($k = 2$) is $d(1, 2) = 1$, whereas the distance to a 5-star review is $d(1, 5) = 4$.

*Emotion Datasets (Empathetic Dialogues, GoEmotions, CancerEMO):* For emotion datasets, we map the target classes onto Plutchik's wheel of emotions. We define the semantic distance $d(y, k)$ as the shortest path distance (i.e., the number of adjacent steps) between emotion $y$ and emotion $k$ along the wheel. For instance, the distance between *Joy* and *Trust* is 1, while the distance between *Joy* and its polar opposite, *Sadness*, is 4. For secondary emotions in larger taxonomies (e.g., GoEmotions), distances are computed based on their primary emotion roots.

*Probability Formulation:* Given a target overall noise rate $\rho \in (0, 1)$ (e.g., $\rho = 0.2$ in our primary experiments), the probability of an instance retaining its true label remains $P(\tilde{y} = y \mid y) = 1 - \rho$.

For any incorrect candidate class $k \neq y$, the probability of the label being flipped to $k$ is calculated by normalizing the exponential decay of the semantic distance across all possible incorrect classes:

$$P(\tilde{y} = k \mid y) = \rho \cdot \frac{\exp(-\gamma \cdot d(y, k))}{\sum_{j \neq y} \exp(-\gamma \cdot d(y, j))} \qquad (10)$$

where $\gamma > 0$ is a scaling parameter that controls the concentration of the semantic bias. A higher $\gamma$ restricts the injected noise almost exclusively to the most immediate neighboring classes, simulating highly nuanced human annotator ambiguity. In our experiments, we set $\gamma = 1.0$ to ensure a strong bias toward semantic neighbors while still allowing a non-zero probability for distant classes.

## H    NOISE CORRECTION COMPARISON

Furthermore, we compare LANE against recent advancements in noisy label correction, specifically the interpolation-based approach by Xu et al. (2025). As shown in Table 11, LANE outperforms this recent state-of-the-art method on 4 out of 5 evaluated datasets, demonstrating its strong capability in mitigating the effects of label noise.

| Method | Emph | GoEmo | RCV1 | SST-5 | Yelp |
|---|---|---|---|---|---|
| LANE | **60.8** | **66.5** | **59.3** | 58.9 | **69.2** |
| Xu et al. (2025) | 60.2 | 66.4 | 58.7 | **59.2** | 68.5 |

Table 11: Performance comparison of LANE against the recent approach by Xu et al. (2025).

