# OpenReview forum: "LANE: Label-Aware Noise Elimination for Fine-Grained Text Classification"
_ICLR.cc/2026/Conference — ICLR 2026 Poster_

### Official Review · Reviewer_B3Cs · 2025-10-29

**Soundness:** 3
**Presentation:** 3
**Contribution:** 3
**Rating:** 6
**Confidence:** 3

**Summary:**

This paper introduces LANE, a training framework designed to enhance model robustness under noisy labels. LANE dynamically adjusts the contribution of training examples by reducing the weight of samples likely to be mislabeled. To identify such samples, it leverages training and semantic relationships. Extensive experiments across multiple fine-grained text classification tasks demonstrate consistent gains, showcasing LANE’s effectiveness in handling label noise.

**Strengths:**

The reviewer notes the following strengths:
- The paper clearly articulates the limitations of prior methods and justifies the need for LANE.
- The proposed LANE methodology is intuitive and technically sound.
- The authors provide extensive experiments on multipole fine-grained text classification datasets that demonstrate consistent and meaningful improvements over multiple baselines.
- The authors also includes a large set of ablations alongside thoughtful exploration of weight distributions that help provide insight into the underlying methodology.

**Weaknesses:**

From the reviewer's perspective, LANE relies heavily on the foundational capabilities of the underlying model. Therefore additional evaluations on language models beyond BERT would add to the real-world applicability of LANE. But in general, the reviewer finds the overall methodology to be sane and showcase valuable improvements.

**Questions:**

See weakness above.

---

> ### Author Response · Authors · 2025-11-21
>
> Thank you very much for your valuable and constructive feedback and comments that help improve the quality of our paper. We are encouraged that you found our LANE approach intuitive and technically sound, and the performance gains—ranging from 2.4% to 4.5% over strong baselines—to be consistent, demonstrating LANE's effectiveness in handling label noise. We address your concern below.
>
> ### Applicability Beyond BERT
> Indeed, LANE is not limited to BERT and we would like to point out two aspects below.
>
> * **Generalizability:** The Label-Aware Margin relies only on logits and embeddings, which are standard outputs for any deep learning model (RoBERTa, DeBERTa, and even ResNet for images—as LANE can be extended to images as well).
> * **Reason for BERT:** We chose BERT as the backbone model to ensure fair comparison with the extensive list of baselines (AUM, DIVIDEMIX, UNICON, etc.) that primarily use BERT as the standard backbone in the Noise Learning literature.
>
> Per your suggestion, we ran experiments with RoBERTa and show the results in the table below. As we can see from the table, LANE with RoBERTa performs similarly and in some cases pushes the performance further compared to LANE with BERT. We will include a discussion on these aspects and these results in the paper. Thank you again for your comment.
>
> | Method | Empath. | GoEmo. | ISEAR | Cancer. | RCV1 | SciHTC | SST-5 | Amazon | Yelp | Yahoo |
> | :--- | :--- | :--- | :--- | :--- | :--- | :--- | :--- | :--- | :--- | :--- |
> | **BASE (BERT)** | 58.5 | 63.6 | 71.5 | 75.8 | 56.8 | 32.5 | 56.3 | 67.5 | 65.9 | 75.4 |
> | **LANE (BERT)** | 60.8 | 66.5 | 74.3 | 78.2 | **59.3** | 34.1 | 58.9 | 69.7 | 69.2 | 78.4 |
> | **BASE (RoBERTa)** | 59.1 | 64.2 | 71.4 | 76.2 | 57.3 | 32.1 | 56.5 | 67.9 | 66.1 | 75.7 |
> | **LANE (RoBERTa)** | **60.9** | **67.2** | **74.5** | **78.3** | 59.1 | **34.6** | **59.1** | **69.6** | **69.3** | **78.5** |

---

### Official Review · Reviewer_5euU · 2025-10-30

**Soundness:** 3
**Presentation:** 3
**Contribution:** 2
**Rating:** 4
**Confidence:** 3

**Summary:**

LANE provides a method for learning with label noise. The authors suggest that label noise could arise from error-prone labelling processes or just genuinely ambiguous labels and they must be weighed differently. The key idea is to assign high weights to hard but clean examples that have high semantic similarity of class labels and down-weigh truly mislabelled examples.

**Strengths:**

* Addresses an important and novel aspect of handling noisy labels - distinguishing between ambiguous or similar labels and erroneous ones
* The method identifies and retains hard but clean examples with higher weight, taking into consideration the semantic similarity of label names
* Extensive empirical analysis has been done comparing LANE with other existing methods

**Weaknesses:**

* Requires additional network training to learn the weights - a computational overhead
* The label-aware supervise contrastive loss could be explained a bit more intuitively, right now, it requires readers to have a strong prior understanding of contrastive learning
* The method is motivated to handle ambiguous or semantically close label noise, but the experiments only consider 20% random noise settings.
* L95 to L99: piling up such a larger number of references without explaining the individual contributions is not helpful at all

**Questions:**

* Have the authors considered the case of class imbalance?
* How much compute power is required to learn the weights?
* Could the authors provide analysis of the learned weights in relation to semantic similarity?
* How would LANE perform for increased label ambiguity and not just random noise?
* Edit suggestions for better readability
1. Algorithm 1 - lines 170-173 could be written more legibly
2. Font too small for all tables
3. Line 245 typo ‘mislabeled’

---

> ### Author Response · Authors · 2025-11-21
>
> Thank you very much for your valuable and constructive feedback and comments that help improve the quality of our paper. We are encouraged that you found our paper to address an important and novel aspect of handling noisy labels. We would like to note that the performance gains—ranging from 2.4% to 4.5% over strong baselines—demonstrate LANE's effectiveness in handling label noise. We address your concerns and questions below.
>
> ### Computational Overhead
> We agree that the computational overhead of training two networks brings additional computational overhead (however, this is a common trait in robust learning methods like Co-teaching and DivideMix, not unique to our LANE approach). We would like to point out two aspects:
> * **Training Value:** The significant F1 gains (up to 4.5% average improvement) justify the training cost for high-stakes fine-grained tasks where annotation quality is low.
> * **Utilization of small networks:** In the LLM era, utilizing two BERT networks is significantly cheaper than any similar-performance (open-source or proprietary) LLM (with tens or even hundreds of billions of parameters). Our approach is trained end-to-end and training the two BERT models is very fast.
>
> ### Label-Aware Contrastive Loss Explanation
> Thank you for your comment. We will explain more intuitively in the paper the label-aware supervised contrastive loss without requiring readers to have a strong prior understanding of contrastive learning. We will rewrite the explanation of Eq. (8) to explicitly state that we are "softening" the contrastive objective based on semantic similarity, which helps reduce overfitting to label noise. To improve clarity, we will also include a diagram of the overall framework.
>
> ### Experiments on Semantic-aware Noise vs. Random Noise
> Thank you for the great comment. We ran more experiments to address this comment. That is, we created a setup where instead of assigning random labels to example (e.g., when we create the 20% noise dataset), we ensure the label flipping probability is higher for the more semantically close labels (that is, for an example where we randomize the label, we sample a semantically similar class more often than a completely random label - so we bias the labels more towards semantically similar labels to capture this type of ambiguity).
>
> For this experiment, we utilize in our comparisons datasets with well-defined class similarities from emotion detection and sentiment analysis. We compare LANE against the best-performing approaches from our main results. As we can see from the table, LANE continues to outperform all the baselines in this setup. We will include these results in the paper.
>
> **Table: Results on Semantic-Aware Noise**
>
> | Method | Empathetic Dialogues | GoEmotions | Yelp | CancerEMO | SST-5 |
> | :--- | :--- | :--- | :--- | :--- | :--- |
> | **BASE** | 50.3 | 57.2 | 64.2 | 73.4 | 53.3 |
> | **AMG** | 52.7 | 60.3 | 66.2 | 74.8 | 54.1 |
> | **AUM** | 51.2 | 61.1 | 66.5 | 74.1 | 52.7 |
> | **LCL** | 53.6 | 60.1 | 65.4 | 73.9 | 53.2 |
> | **DISC** | 52.4 | 60.2 | 64.3 | 74.2 | 53.6 |
> | **LANE** | **54.2** | **61.4** | **67.4** | **75.1** | **54.3** |
>
>
> ### Readability
> We will reorganize the related work section and ensure that we discuss the individual contributions of each work. We will also fix the font size in tables, correct the typo on Line 245, and improve the legibility of Algorithm 1. Thank you for the comments.
>
> ### Response to Questions:
> * **Class Imbalance:** Fine-grained datasets (like RCV1) are inherently imbalanced. LANE handles this implicitly because "rare" classes often have high loss/margins. By focusing on the relative margin (Label-Aware Margin), LANE prevents the model from ignoring minority classes just because they are hard to learn (that is, instead of eliminating examples that appear to be noisy, LANE learns from all examples but assigns weights according to their perceived level of noise). This is a great point. We will add a discussion on this in the paper.
> * **Compute Power:** Training requires approximately 1.8x the compute of a standard BERT training run (due to the auxiliary net), but converges in similar epoch counts. This fits comfortably on a single GPU.
> * **Weight Analysis:** Please see Figure 1 (Page 9) and Table 1 (Page 5) in the paper. These explicitly analyze the learned weights in relation to semantic similarity. Table 1 in the paper shows how the margin adapts when the assigned label is semantically close vs. distant to the prediction.

---

### Official Review · Reviewer_6TRi · 2025-10-30

**Soundness:** 3
**Presentation:** 3
**Contribution:** 3
**Rating:** 6
**Confidence:** 3

**Summary:**

This paper introduces Label-Aware Noise Elimination (LANE), a novel approach to enhance deep learning model robustness against label noise in fine-grained text classification—a task where classes are semantically similar (e.g., distinguishing between subtypes of news articles or product reviews), making label noise particularly detrimental and common. LANE’s core design integrates two complementary signals to identify and downweight noisy training examples: (1) semantic relations between classes and (2) model training dynamics .

**Strengths:**

1. Fine-grained text classification is ubiquitous in real-world applications (e.g., e-commerce product categorization) but uniquely vulnerable to label noise—human annotators often confuse semantically similar classes.
2. LANE’s fusion of dynamic training dynamics (capturing how the model learns an example over time) and static class semantics (capturing inherent class ambiguities) is innovative.
3.  The reported F1 improvements (2.4–4.5%) are meaningful for fine-grained tasks

**Weaknesses:**

1. it is not clear how the supervised contrastive loss helps in Eq (8)? Is there any intuitive illustration?

**Questions:**

see weakness

---

> ### Author Response · Authors · 2025-11-21
>
> Thank you very much for your valuable and constructive feedback and comments that help improve the quality of our paper. We are encouraged that you found our LANE approach innovative and the performance gains—ranging from 2.4% to 4.5% over strong baselines—to be meaningful and significant. We address your concerns and questions below.
>
> ### Intuition of Supervised Contrastive Loss (Eq. 8)
> First, the fixed AUM penalty used in previous work is the same for all annotation errors (no matter how similar or distant an assigned label is from the true-hidden-label). For example, the penalty of an example annotated with "anger" when the true conveyed emotion is "annoyance" is the same as that for an example annotated with "anger" when the true conveyed emotion is "joy". We argue that the penalty should be higher for the latter (when the true and annotated classes are more distant such as "joy" and "anger", respectively).
>
> Second, while we penalize the annotation errors differently (lower penalty for semantically similar classes, and higher otherwise), we do not want the model to learn that similar emotions (e.g., "annoyance" and "anger") are the same. Instead, we want to encourage the model to correctly distinguish between these semantically similar classes and to learn that they are two similar but still different classes. This is achieved through the supervised contrastive loss that pushes the examples from the same class closer together and the examples from different classes (no matter how similar) further apart (to ensure all classes are learned properly).
>
> Eq. (8) uses the weights $w_{x_i,y_s}$ (derived from the auxiliary network) to modulate the proper learning of fine-grained classes and specifically learn representations that bring closer examples from the same class and push further apart those from different classes. We will add a conceptual figure in the paper to illustrate the overall LANE architecture. Thank you again for your comment.

---

### Official Review · Reviewer_xwV9 · 2025-11-01

**Soundness:** 3
**Presentation:** 3
**Contribution:** 3
**Rating:** 4
**Confidence:** 3

**Summary:**

This paper proposes Label-Aware Noise Elimination (LANE), a method that introduces a novel metric called label-aware margin. By incorporating inter-class semantic similarities and model training dynamics, LANE quantifies the label noise level of each training sample. It dynamically down-weights suspected noisy samples, thereby mitigating the harmful effects of noisy labels while retaining all training samples (including hard but clean ones). Experiments across ten text classification datasets demonstrate LANE's effectiveness under varying noise levels.

**Strengths:**

1. Clear Motivation: The paper effectively identifies two key limitations in existing methods: (1) valuable samples below a low AUM threshold are unnecessarily removed from the training set, and (2) AUM computation treats labels independently, ignoring semantic similarities between them. The authors propose LANE to address these specific shortcomings.
2. Clear Methodology: The theoretical analysis and step-by-step description of the proposed method are clear and well-presented.
3. Extensive Experiments: The comparative experiments are comprehensive, covering numerous datasets spanning most text classification tasks. The proposed LANE method achieves near-universal improvements over baselines.

**Weaknesses:**

1. Several works on noisy label learning emerged in 2025. As a submission to a 2026 conference, the lack of comparison with new methods such as [1], [2], and [3] detracts from the overall novelty of the work.
2. The approach requires training two BERT models for text classification, which is inefficient. Furthermore, while LLM-related experiments are included, the LLMs rely solely on context. The work could be strengthened by incorporating parameter-efficient fine-tuning techniques like LoRA or integrating noisy label learning methods specifically designed for LLMs, such as [3], to enhance persuasiveness.
3. The Related Work section is disorganized. Noisy label learning encompasses various approaches, but the authors resort to simple listing. This issue is also reflected in the main experiments (Table 2), which lack intuitive grouping. It is recommended to group and present comparisons by methodology type.
4. This paper lacks the reproducibility statement required by ICLR.

References:

[1] Pan et al., Enhanced Sample Selection with Confidence Tracking: Identifying Correctly Labeled yet Hard-to-Learn Samples in Noisy Data, AAAI, 2025.

[2] Xu et al., Revisiting Interpolation for Noisy Label Correction, AAAI 2025

[3] Ye et al., Calibrating Pre-trained Language Classifiers on LLM-generated Noisy Labels via Iterative Refinement, SIGKDD 2025.

**Questions:**

1. The experiments use BERT as the base model. Is LANE compatible with other architectures and large language models?
2. Why wasn't PLF compared in the 20% noise setting experiments?
3. The complexity of this work is not low. Why not include a main framework diagram? This would make the overall workflow and contributions of the work clearer.

---

> ### Author Response · Authors · 2025-11-21
>
> Thank you very much for your valuable and constructive feedback and comments that help improve the quality of our paper. We are encouraged that you found our experiments to be comprehensive and the performance improvements—ranging from 2.4% to 4.5% over strong baselines—to be meaningful and significant. We address the concerns and questions below.
>
> ### Comparison with Recent Works
> Thank you for pointing out the recent related works. We ran experiments on several datasets with the approach by Xu et al, (AAAI 2025) since, like us, this approach presents a way to deal with / correct noisy labels (and it appears to be the most similar to us). We report the results comparing LANE with this approach in the table below and we will include these results in the paper.
>
> As we can see from the table, LANE outperforms Xu et al (2025)'s approach on 4 out of 5 datasets. We will cite all the suggested references and discuss them in our paper.
>
> | Method | Empathetic Dialogues | GoEmotions | RCV1 | SST-5 | Yelp |
> | :--- | :--- | :--- | :--- | :--- | :--- |
> | **LANE** | **60.8** | **66.5** | **59.3** | 58.9 | **69.2** |
> | [2] Xu et al. | 60.2 | 66.4 | 58.7 | **59.2** | 68.5 |
>
>
> ### Efficiency (Two BERT models)
> We agree that the computational overhead of training two networks brings additional computational overhead (however, this is a common trait in robust learning methods like Co-teaching and Divide Mix, not unique to our LANE approach). We would like to point out two aspects:
>
> * **Training Value:** The significant F1 gains (up to 4.5% average improvement) justify the training cost for high-stakes fine-grained tasks where annotation quality is low.
> * **Utilization of small networks:** In the LLM era, utilizing two BERT networks is significantly cheaper than any similar-performance (open-source or proprietary) LLM (with tens or even hundreds of billions of parameters). Our approach is trained end-to-end and training the two BERT models is very fast.
>
> ### LLM Experiments
> We appreciate the suggestion regarding LoRA.
>
> * We currently compare in the paper against LLMs (ChatGPT and Llama-2) in Table 7 (Appendix D) in a few-shot setting. LANE significantly outperforms Llama-2 (e.g., +18.7% on RCV1) and ChatGPT on most datasets.
>
> * Fine-tuning LLMs even when incorporating parameter-efficient fine-tuning techniques like LoRA is computationally more inefficient than training two BERT models end-to-end. We will add a discussion on this in the paper and will cite the suggested related work.
>
> * It is also more common or desirable to just prompt an LLM (with or without in-context examples) rather than fine-tuning the LLM (hence, our choice for the comparison with the LLM in-context learning).
>
> ### Reorganization & Reproducibility
> We will reorganize the Related Work section by methodology (e.g., Sample Selection, Noise Transition Matrix, Contrastive Learning), as suggested, and will ensure that we will discuss all related works to highlight their contributions and differences between LANE and the previous works. We confirm that the code will be released (we will reference it in the abstract/conclusion) and we will add a formal Reproducibility Statement in the final version of the paper.
>
> ### Questions:
>
> **1. Compatibility with other architectures.**
> Yes, LANE is compatible with other architectures, e.g., RoBERTa, DeBERTa, etc. Please see the response to Reviewer B3Cs for this question where we show additional experiments with RoBERTa.
>
> **2. PLF comparison in the 20% noise setting.**
> Thank you for pointing out this experiment that we overlooked! We attach the results below, showing that LANE consistently outperforms PLF on 20% noise experiments as well on all datasets. We will include the results in the paper.
>
> **Table: Results on 20% Noise (LANE vs. PLF)**
>
> | Method | Empath. | GoEmo. | ISEAR | Cancer. | RCV1 | SciHTC | SST-5 | Amazon | Yelp | Yahoo |
> | :--- | :--- | :--- | :--- | :--- | :--- | :--- | :--- | :--- | :--- | :--- |
> | **PLF** | 12.5 | 16.7 | 36.8 | 50.1 | 48.2 | 29.7 | 51.9 | 62.1 | 64.7 | 68.1 |
> | **LANE** | **15.9** | **24.3** | **40.4** | **52.5** | **49.4** | **30.5** | **53.1** | **63.1** | **65.2** | **68.9** |
>
>
> **3. Framework diagram.**
> We will include a main framework diagram for clarity of the approach.

---

### Author Response · Authors · 2025-12-04

We thank the AC for handling our paper. We also thank our Reviewers for their constructive feedback and comments. We are encouraged that the Reviewers found our LANE approach to be "innovative" and "technically sound". The Reviewers highlighted our "extensive" and "comprehensive" experiments and agreed that the performance improvements (ranging from 2.4% to 4.5% over strong baselines) are meaningful for fine-grained text classification. We summarize below the main points from our response to Reviewers.

In our response, we have addressed the Reviewers' concerns as follows:

### 1. Comparison with Recent SOTA (Reviewer xwV9)
Reviewer xwV9 pointed out missing recent works from 2025. In our response, we conducted new experiments comparing LANE against Xu et al. (AAAI 2025). The results demonstrate that LANE outperforms this recent approach on 4 out of 5 datasets, achieving notable gains. We also added a comparison against the Peer Loss approach (PLF) in the 20% noise setting, where LANE consistently outperformed the baseline across all datasets, showing significant improvements.

### 2. Generalizability and Model Architecture (Reviewer B3Cs)
Addressing the concern regarding LANE's reliance on BERT, we conducted additional experiments using RoBERTa. The results show that LANE is model-agnostic and achieves similar or superior performance gains with the RoBERTa backbone, in some cases pushing performance even further than the BERT implementation.

### 3. Robustness to Semantic Noise (Reviewer 5euU)
To address Reviewer 5euU's question on whether LANE handles only random noise, we implemented a new "Semantic-Aware Noise" experimental setup. In this setup, label flipping probability is higher for semantically similar classes. Our new results confirm that LANE continues to outperform all baselines (including strong methods like AUM, LCL, and DISC) in this challenging, realistic scenario.

### 4. Clarifications on Methodology and Efficiency (Reviewers xwV9, 6TRI, 5euU)

* **Class Imbalance:** We addressed the query regarding class imbalance, clarifying that LANE handles this implicitly. Since "rare" classes often exhibit high margins, LANE's use of relative Label-Aware Margin prevents the model from neglecting minority classes by assigning them appropriate weights rather than eliminating them as "noise".
* **Efficiency:** We clarified that while training two networks adds overhead, it remains significantly more efficient than fine-tuning Large Language Models (LLMs) while providing substantial F1 gains.
* **Contrastive Loss (Eq. 8):** We provided a detailed intuition for the label-aware supervised contrastive loss, explaining how it imposes higher penalties for distant classes while maintaining distinction between semantically similar classes.

We believe these additional experiments and clarifications significantly strengthen the paper and demonstrate LANE's robustness and novelty.

---

### Meta-Review · Area_Chair_T1eo · 2026-01-07

**Summary:**

This paper proposes LANE (Label-Aware Noise Elimination) for fine-grained text classification under label noise. The key idea is a label-aware margin that incorporates semantic relationships between classes (to distinguish “ambiguous but plausible” labels from truly wrong ones) together with training dynamics to down-weight suspected noisy instances rather than hard-filtering them. Across a broad suite of datasets and noise settings, LANE reports consistent gains over strong baselines, with average improvements around 2.4% F1 on manually annotated datasets and 4.5% F1 under higher noise. The revision/rebuttal adds comparisons to recent 2025 work, extends to RoBERTa, and includes a semantic-aware (non-uniform) noise setup, all of which strengthen the case.

**Reviewer Concerns:**

Addressed concerns
- Missing recent SOTA comparisons (xwV9): Authors added experiments vs Xu et al. (AAAI 2025) and show LANE winning on 4/5 datasets; they also added PLF results for the 20% noise setting, where LANE consistently outperforms PLF across all datasets. This directly addresses novelty/positioning concerns.
- Generalizability beyond BERT (B3Cs): Authors provide RoBERTa experiments showing similar or slightly better gains, supporting that LANE is not backbone-specific.
- Noise model realism (5euU): Authors add a semantic-aware noise setting (higher flip probability among semantically similar labels) and LANE remains best among strong baselines. This is important because it better matches fine-grained label ambiguity.
- Clarity on supervised contrastive component (6TRi, 5euU): Rebuttal gives a reasonable intuition: modulate penalties by semantic distance while still separating semantically close classes; they commit to adding a conceptual figure/diagram.
- Class imbalance + compute (5euU): Authors respond that LANE implicitly handles imbalance via relative label-aware margin and estimate compute at ~1.8× a standard run; “fits on a single GPU” is plausible for BERT/RoBERTa-scale.

Remaining concerns (minor / revision-level)
- Efficiency / two-network overhead (xwV9, 5euU): Still an overhead, but the paper is squarely in robust learning; the rebuttal frames it well and quantifies overhead. I view this as acceptable for ICLR.
- Paper organization / related work grouping + reproducibility statement (xwV9, 5euU): Authors commit to reorganizing and adding the required reproducibility statement and a main framework diagram. This should be required in the final version, but not a reason to reject given the strength of evidence.

**Reviewer Scores:**

Based on the rebuttal:
- xwV9 (original 4): likely moves to 5–6 (major concerns about missing 2025 comparisons and PLF at 20% noise were directly addressed; also commits to reproducibility + reorg).
- 5euU (original 4): likely moves to 5–6 (semantic-aware noise experiment + compute + imbalance addressed; contrastive intuition improved).
- 6TRi (original 6): stays 6 (main question answered).
- B3Cs (original 6): stays 6 (RoBERTa experiment added).

Overall, the trajectory is upward with rebuttal.

---

### Decision · Program_Chairs · 2026-01-26

Accept (Poster)